# Generalized Leverage Score Sampling for Neural Networks[*]

**Jason D. Lee**[†]     **Ruoqi Shen**[‡]     **Zhao Song**[§]     **Mengdi Wang**[¶]     **Zheng Yu**[‖]

## Abstract

Leverage score sampling is a powerful technique that originates from theoretical computer science, which can be used to speed up a large number of fundamental questions, e.g. linear regression, linear programming, semi-definite programming, cutting plane method, graph sparsification, maximum matching and max-flow. Recently, it has been shown that leverage score sampling helps to accelerate kernel methods [Avron, Kapralov, Musco, Musco, Velingker and Zandieh 17].

In this work, we generalize the results in [Avron, Kapralov, Musco, Musco, Velingker and Zandieh 17] to a broader class of kernels. We further bring the leverage score sampling into the field of deep learning theory.

- We show the connection between the initialization for neural network training and approximating the neural tangent kernel with random features.
- We prove the equivalence between regularized neural network and neural tangent kernel ridge regression under the initialization of both classical random Gaussian and leverage score sampling.

## 1 Introduction

Kernel method is one of the most common techniques in various machine learning problems. One classical application is the kernel ridge regression (KRR). Given training data $X = [x_1, \cdots, x_n]^\top \in \mathbb{R}^{n \times d}$, corresponding labels $Y = [y_1, \cdots, y_n] \in \mathbb{R}^n$ and regularization parameter $\lambda > 0$, the output estimate of KRR for any given input $z$ can be written as:

$$f(z) = \mathsf{K}(z, X)^\top (K + \lambda I_n)^{-1} Y, \tag{1}$$

where $\mathsf{K}(\cdot, \cdot)$ denotes the kernel function and $K \in \mathbb{R}^{n \times n}$ denotes the kernel matrix.

Despite being powerful and well-understood, the kernel ridge regression suffers from the costly computation when dealing with large datasets, since generally implementation of Eq. (1) requires $O(n^3)$ running time. Therefore, intensive research have been dedicated to the scalable methods for KRR [Bac13, AM15, ZDW15, ACW17, MM17, ZNV+20]. One of the most popular approach is the random Fourier features sampling originally proposed by [RR08] for shift-invariant kernels. They construct a finite dimensional random feature vector $\phi : \mathbb{R}^d \to \mathbb{C}^s$ through sampling that approximates the kernel function $\mathsf{K}(x, z) \approx \phi(x)^* \phi(z)$ for data $x, z \in \mathbb{R}^d$. The random feature helps approximately solves KRR in $O(ns^2 + n^2)$ running time, which improves the computational

---

[*]The authors would like to thank Michael Kapralov for suggestion of this topic. The full version of this paper is available at `https://arxiv.org/pdf/2009.09829.pdf`

[†]`jasonlee@princeton.edu` Princeton University. Work done while visiting Institute for Advanced Study.

[‡]`shenr3@cs.washington.edu` University of Washington. Work done while visiting Institute for Advanced Study.

[§]`zhaos@ias.edu` Columbia University, Princeton University and Institute for Advanced Study.

[¶]`mengdiw@princeton.edu` Princeton University.

[‖]`zhengy@princeton.edu` Princeton University.

cost if $s \ll n$. The work [AKM+17] advanced this result by introducing the leverage score sampling to take the regularization term into consideration.

In this work, we follow the the approach in [AKM+17] and naturally generalize the result to a broader class of kernels, which is of the form

$$\mathsf{K}(x, z) = \mathop{\mathbb{E}}_{w \sim p} [\phi(x, w)^\top \phi(z, w)],$$

where $\phi : \mathbb{R}^d \times \mathbb{R}^{d_1} \to \mathbb{R}^{d_2}$ is a finite dimensional vector and $p : \mathbb{R}^{d_1} \to \mathbb{R}_{\geq 0}$ is a probability distribution. We apply the leverage score sampling technique in this generalized case to obtain a tighter upper-bound on the dimension of random features.

The most important contribution of this work is to introduce the leverage score theory into the field of neural network training. Over the last two years, there is a long line of over-parametrization theory works on the convergence results of deep neural network [LL18, DZPS19, AZLS19b, AZLS19a, DLL+19, ADH+19b, ADH+19a, SY19, BPSW20], all of which either explicitly or implicitly use the property of neural tangent kernel [JGH18]. However, most of those results focus on neural network training without regularization, while in practice regularization (which is originated from classical machine learning) has been widely used in training deep neural network. Therefore, in this work we rigorously build the equivalence between training a ReLU deep neural network with $\ell_2$ regularization and neural tangent kernel ridge regression. We observe that the initialization of training neural network corresponds to approximating the neural tangent kernel with random features, whose dimension is proportional to the width of the network. Thus, it motivates us to bring the leverage score sampling theory into the neural network training. We present a new equivalence between neural net and kernel ridge regression under the initialization using leverage score sampling, which potentially improves previous equivalence upon the upper-bound of network width needed.

We summarize our main results and contribution as following:

- Generalize the leverage score sampling theory for kernel ridge regression to a broader class of kernels.
- Connect the leverage score sampling theory with neural network training.
- Theoretically prove the equivalence between training regularized neural network and kernel ridge regression under both random Gaussian initialization and leverage score sampling initialization.

## 2 Related work

**Leverage scores**     Given a $m \times n$ matrix $A$. Let $a_i^\top$ be the $i$-th rows of $A$ and the leverage score of the $i$-th row of $A$ is $\sigma_i(A) = a_i^\top (A^\top A)^\dagger a_i$. A row's leverage score measures how important it is in composing the row space of $A$. If a row has a component orthogonal to all other rows, its leverage score is 1. Removing it would decrease the rank of $A$, completely changing its row space. The coherence of $A$ is $\|\sigma(A)\|_\infty$. If $A$ has low coherence, no particular row is especially important. If $A$ has high coherence, it contains at least one row whose removal would significantly affect the composition of $A$'s row space.

Leverage score is a fundamental concept in graph problems and numerical linear algebra. There are many works on how to approximate leverage scores [SS11, DMIMW12, CW13, NN13] or more general version of leverages, e.g. Lewis weights [Lew78, BLM89, CP15]. From graph perspective, it has been applied to maximum matching [BLN+20, LSZ20], max-flow [DS08, Mad13, Mad16, LS20b, LS20a], generate random spanning trees [Sch18], and sparsify graphs [SS11]. From matrix perspective, it has been used to give matrix CUR decomposition [BW14, SWZ17, SWZ19] and tensor CURT decomposition [SWZ19]. From optimization perspective, it has been used to approximate the John Ellipsoid [CCLY19], linear programming [LS14, BLSS20, JSWZ20], semi-definite programming [JKL+20], and cutting plane methods [Vai89, LSW15, JLSW20].

**Kernel methods**     Kernel methods can be thought of as instance-based learners: rather than learning some fixed set of parameters corresponding to the features of their inputs, they instead "remember" the $i$-th training example $(x_i, y_i)$ and learn for it a corresponding weight $w_i$. Prediction for unlabeled inputs, i.e., those not in the training set, is treated by the application of similarity function $\mathsf{K}$, called a kernel, between the unlabeled input $x'$ and each of the training inputs $x_i$.

There are three lines of works that are closely related to our work. First, our work is highly related to the recent discoveries of the connection between deep learning and kernels [DFS16, Dan17, JGH18, CB18]. Second, our work is closely related to development of connection between leverage score and kernels [RR08, CW17, CMM17, MW17b, MW17a, LTOS18, AKM$^+$17, AKM$^+$19, ACSS20]. Third, our work is related to kernel ridge regression [Bac13, AM15, ZDW15, ACW17, MM17, ZNV$^+$20].

**Convergence of neural network**   There is a long line of work studying the convergence of neural network with random input assumptions [BG17, Tia17, ZSJ$^+$17, Sol17, LY17, ZSD17, DLT$^+$18, GLM18, BJW19]. For a quite while, it is not known to remove the randomness assumption from the input data points. Recently, there is a large number of work studying the convergence of neural network in the over-parametrization regime [LL18, DZPS19, AZLS19b, AZLS19a, DLL$^+$19, ADH$^+$19b, ADH$^+$19a, SY19, BPSW20]. These results don't need to assume that input data points are random, and only require some much weaker assumption which is called "data-separable". Mathematically, it says for any two input data points $x_i$ and $x_j$, we have $\|x_i - x_j\|_2 \geq \delta$. Sufficiently wide neural network requires the width $m$ to be at least $\mathrm{poly}(n, d, L, 1/\delta)$, where $n$ is the number of input data points, $d$ is the dimension of input data point, $L$ is the number of layers.

**Continuous Fourier transform**   The continuous Fourier transform is defined as a problem [JLS20] where you take samples $f(t_1), \cdots, f(t_m)$ from the time domain $f(t) := \sum_{j=1}^n v_j e^{2\pi \mathbf{i} \langle x_j, t \rangle}$, and try to reconstruct function $f : \mathbb{R}^d \to \mathbb{C}$ or even recover $\{(v_j, x_j)\} \in \mathbb{C} \times \mathbb{R}^d$. The data separation connects to the sparse Fourier transform in the continuous domain. We can view the $n$ input data points [LL18, AZLS19b, AZLS19a] as $n$ frequencies in the Fourier transform [Moi15, PS15]. The separation of the data set is equivalent to the gap of the frequency set ($\min_{i \neq j} \|x_i - x_j\|_2 \geq \delta$). In the continuous Fourier transform, there are two families of algorithms: one requires to know the frequency gap [Moi15, PS15, CM20, JLS20] and the other doesn't [CKPS16]. However, in the over-parameterized neural network training, all the existing work requires a gap for the data points.

**Notations**   We use $\mathbf{i}$ to denote $\sqrt{-1}$. For vector $x$, we use $\|x\|_2$ to denote the $\ell_2$ norm of $x$. For matrix $A$, we use $\|A\|$ to denote the spectral norm of $A$ and $\|A\|_F$ to denote the Frobenius norm of $A$. For matrix $A$ and $B$, we use $A \preceq B$ to denote that $B - A$ is positive semi-definite. For a square matrix, we use $\mathrm{tr}[A]$ to denote the trace of $A$. We use $A^{-1}$ to denote the true inverse of an invertible matrix. We use $A^\dagger$ to denote the pseudo-inverse of matrix $A$. We use $A^\top$ to denote the transpose of matrix $A$.

## 3   Main results

In this section, we state our results. In Section 3.1, we consider the large-scale kernel ridge regression (KRR) problem. We generalize the Fourier transform result [AKM$^+$17] of accelerating the running time of solving KRR using the tool of leverage score sampling to a broader class of kernels. In Section 3.2, we discuss the interesting application of leverage score sampling for training deep learning models due to the connection between regularized neural nets and kernel ridge regression.

### 3.1   Kernel approximation with leverage score sampling

In this section, we generalize the leverage score theory in [AKM$^+$17], which analyzes the number of random features needed to approximate kernel matrix under leverage score sampling regime for the kernel ridge regression task. In the next a few paragraphs, we briefly review the settings of classical kernel ridge regression.

Given training data given training data matrix $X = [x_1, \cdots, x_n]^\top \in \mathbb{R}^{n \times d}$, corresponding labels $Y = [y_1, \cdots, y_n]^\top \in \mathbb{R}^n$ and feature map $\phi : \mathbb{R}^d \to \mathcal{F}$, a classical kernel ridge regression problem

can be written as[7]

$$\min_{\beta} \frac{1}{2}\|Y - \phi(X)^{\top}\beta\|_2^2 + \frac{1}{2}\lambda\|\beta\|_2^2$$

where $\lambda > 0$ is the regularization parameter. By introducing the corresponding kernel function $\mathsf{K}(x,z) = \langle \phi(x), \phi(z) \rangle$ for any data $x, z \in \mathbb{R}^d$, the output estimate of the kernel ridge regression for any data $x \in \mathbb{R}^d$ can be denoted as $f^*(x) = \mathsf{K}(x, X)^{\top}\alpha$, where $\alpha \in \mathbb{R}^n$ is the solution to

$$(K + \lambda I_n)\alpha = Y.$$

Here $K \in \mathbb{R}^{n \times n}$ is the kernel matrix with $K_{i,j} = \mathsf{K}(x_i, x_j), \forall i, j \in [n] \times [n]$.

Note a direct computation involves $(K + \lambda I_n)^{-1}$, whose $O(n^3)$ running time can be fairly large in tasks like neural network due to the large number of training data. Therefore, we hope to construct feature map $\phi : \mathbb{R}^d \to \mathbb{R}^s$, such that the new feature approximates the kernel matrix well in the sense of

$$(1 - \epsilon) \cdot (K + \lambda I_n) \preceq \Phi\Phi^{\top} + \lambda I_n \preceq (1 + \epsilon) \cdot (K + \lambda I_n), \tag{2}$$

where $\epsilon \in (0, 1)$ is small and $\Phi = [\phi(x_1), \cdots, \phi(x_n)]^{\top} \in \mathbb{R}^{n \times s}$. Then by Woodbury matrix equality, we can approximate the solution by $u^*(z) = \phi(z)^{\top}(\Phi^{\top}\Phi + \lambda I_s)^{-1}\Phi^{\top}Y$, which can be computed in $O(ns^2 + n^2)$ time. In the case $s = o(n)$, computational cost can be saved.

In this work, we consider a generalized setting of [AKM$^+$17] as a kernel ridge regression problem with positive definite kernel matrix $\mathsf{K} : \mathbb{R}^d \times \mathbb{R}^d \to \mathbb{R}$ of the form

$$\mathsf{K}(x, z) = \mathop{\mathbb{E}}_{w \sim p}[\phi(x, w)^{\top}\phi(z, w)], \tag{3}$$

where $\phi : \mathbb{R}^d \times \mathbb{R}^{d_1} \to \mathbb{R}^{d_2}$ denotes a finite dimensional vector and $p : \mathbb{R}^{d_1} \to \mathbb{R}_{\geq 0}$ denotes a probability density function.

Due to the regularization $\lambda > 0$ in this setting, instead of constructing the feature map directly from the distribution $q$, we consider the following ridge leveraged distribution:

**Definition 3.1** (Ridge leverage function). *Given data $x_1, \cdots, x_n \in \mathbb{R}^d$ and parameter $\lambda > 0$, we define the ridge leverage function as*

$$q_{\lambda}(w) = p(w) \cdot \mathrm{tr}[\Phi(w)^{\top}(K + \lambda I_n)^{-1}\Phi(w)],$$

*where $p(\cdot)$, $\phi$ are defined in Eq. (3), and $\Phi(w) = [\phi(x_1, w)^{\top}, \cdots, \phi(x_n, w)^{\top}]^{\top} \in \mathbb{R}^{n \times d_2}$. Further, we define statistical dimension $s_{\lambda}(K)$ as*

$$s_{\lambda}(K) = \int q_{\lambda}(w)\mathrm{d}w = \mathrm{tr}[(K + \lambda I_n)^{-1}K]. \tag{4}$$

The leverage score sampling distribution $q_{\lambda}/s_{\lambda}(K)$ takes the regularization term into consideration and achieves Eq. (2) using the following modified random features vector:

**Definition 3.2** (Modified random features). *Given any probability density function $q(\cdot)$ whose support includes that of $p(\cdot)$. Given $m$ vectors $w_1, \cdots, w_m \in \mathbb{R}^{d_1}$, we define modified random features $\overline{\Psi} \in \mathbb{R}^{n \times m d_2}$ as $\overline{\Psi} := [\overline{\varphi}(x_1), \cdots, \overline{\varphi}(x_n)]^{\top}$, where*

$$\overline{\varphi}(x) = \frac{1}{\sqrt{m}} \left[ \frac{\sqrt{p(w_1)}}{\sqrt{q(w_1)}}\phi(x, w_1)^{\top}, \cdots, \frac{\sqrt{p(w_m)}}{\sqrt{q(w_m)}}\phi(x, w_m)^{\top} \right]^{\top}.$$

Now we are ready to present our result.

**Theorem 3.3** (Kernel approximation with leverage score sampling, generalization of Lemma 8 in [AKM$^+$17]). *Given parameter $\lambda \in (0, \|K\|)$. Let $q_{\lambda} : \mathbb{R}^{d_1} \to \mathbb{R}_{\geq 0}$ be the leverage score defined in Definition 3.1. Let $\widetilde{q}_{\lambda} : \mathbb{R}^{d_1} \to \mathbb{R}$ be any measurable function such that $\widetilde{q}_{\lambda}(w) \geq q_{\lambda}(w)$ holds*

*for all $w \in \mathbb{R}^{d_1}$. Assume $s_{\widetilde{q}_\lambda} = \int_{\mathbb{R}^{d_1}} \widetilde{q}_\lambda(w)\mathrm{d}w$ is finite. Let $\overline{q}_\lambda(w) = \widetilde{q}_\lambda(w)/s_{\widetilde{q}_\lambda}$. Given any accuracy parameter $\epsilon \in (0, 1/2)$ and failure probability $\delta \in (0, 1)$. Let $w_1, \cdots, w_m \in \mathbb{R}^d$ denote $m$ samples draw independently from the distribution associated with the density $\overline{q}_\lambda(\cdot)$, and construct the modified random features $\overline{\Psi} \in \mathbb{R}^{n \times md_2}$ as in Definition 3.2 with $q = \overline{q}_\lambda$. Let $s_\lambda(K)$ be the statistical dimension defined in (4). If $m \geq 3\epsilon^{-2} s_{\widetilde{q}_\lambda} \ln(16 s_{\widetilde{q}_\lambda} \cdot s_\lambda(K)/\delta)$, then we have*

$$(1 - \epsilon) \cdot (K + \lambda I_n) \preceq \overline{\Psi\Psi}^\top + \lambda I_n \preceq (1 + \epsilon) \cdot (K + \lambda I_n) \tag{5}$$

*holds with probability at least $1 - \delta$.*

**Remark 3.4.** *Above results can be generalized to the complex domain $\mathbb{C}$. Note for the random Fourier feature case discussed in [AKM+17], we have $d_1 = d$, $d_2 = 1$, $\phi(x, w) = e^{-2\pi\mathbf{i}w^\top x} \in \mathbb{C}$ and $p(\cdot)$ denotes the Fourier transform density distribution, which is a special case in our setting.*

## 3.2 Application in training regularized neural network

In this section, we consider the application of leverage score sampling in training $\ell_2$ regularized neural networks.

Past literature such as [DZPS19],[ADH+19a] have already witnessed the equivalence between training a neural network and solving a kernel regression problem in a broad class of network models. In this work, we first generalize this result to the regularization case, where we connect regularized neural network with kernel ridge regression. Then we apply the above discussed the leverage score sampling theory for KRR to the task of training neural nets.

### 3.2.1 Equivalence I, training with random Gaussian initialization

To illustrate the idea, we consider a simple model two layer neural network with ReLU activation function as in [DZPS19, SY19][8].

$$f_{\mathrm{nn}}(W, a, x) = \frac{1}{\sqrt{m}} \sum_{r=1}^{m} a_r \sigma(w_r^\top x) \in \mathbb{R},$$

where $x \in \mathbb{R}^d$ is the input, $w_r \in \mathbb{R}^d$, $r \in [m]$ is the weight vector of the first layer, $W = [w_1, \cdots, w_m] \in \mathbb{R}^{d \times m}$, $a_r \in \mathbb{R}$, $r \in [m]$ is the output weight, $a = [a_1, \cdots, a_m]^\top$ and $\sigma(\cdot)$ is the ReLU activation function: $\sigma(z) = \max\{0, z\}$.

Here we consider only training the first layer $W$ with fixed $a$, so we also write $f_{\mathrm{nn}}(W, x) = f_{\mathrm{nn}}(W, a, x)$. Again, given training data matrix $X = [x_1, \cdots, x_n]^\top \in \mathbb{R}^{n \times d}$ and labels $Y = [y_1, \cdots, y_n]^\top \in \mathbb{R}^n$, we denote $f_{\mathrm{nn}}(W, X) = [f_{\mathrm{nn}}(W, x_1), \cdots, f_{\mathrm{nn}}(W, x_n)]^\top \in \mathbb{R}^n$. We formally define training neural network with $\ell_2$ regularization as follows:

**Definition 3.5** (Training neural network with regularization). *Let $\kappa \in (0, 1]$ be a small multiplier[9]. Let $\lambda \in (0, 1)$ be the regularization parameter. We initialize the network as $a_r \overset{i.i.d.}{\sim} \mathrm{unif}[\{-1, 1\}]$ and $w_r(0) \overset{i.i.d.}{\sim} \mathcal{N}(0, I_d)$. Then we consider solving the following optimization problem using gradient descent:*

$$\min_W \frac{1}{2}\|Y - \kappa f_{\mathrm{nn}}(W, X)\|_2^2 + \frac{1}{2}\lambda\|W\|_F^2. \tag{6}$$

*Let $w_r(t), r \in [m]$ be the network weight at iteration $t$. We denote the training data predictor at iteration $t$ as $u_{\mathrm{nn}}(t) = \kappa f_{\mathrm{nn}}(W(t), X) \in \mathbb{R}^n$. Further, given any test data $x_{\mathrm{test}} \in \mathbb{R}^d$, we denote $u_{\mathrm{nn,test}}(t) = \kappa f_{\mathrm{nn}}(W(t), x_{\mathrm{test}}) \in \mathbb{R}$ as the test data predictor at iteration $t$.*

On the other hand, we consider the following neural tangent kernel ridge regression problem:

$$\min_\beta \frac{1}{2}\|Y - \kappa f_{\mathrm{ntk}}(\beta, X)\|_2^2 + \frac{1}{2}\lambda\|\beta\|_2^2, \tag{7}$$

where $\kappa, \lambda$ are the same parameters as in Eq. (6), $f_{\mathrm{ntk}}(\beta, x) = \Phi(x)^\top \beta \in \mathbb{R}$ and $f_{\mathrm{ntk}}(\beta, X) = [f_{\mathrm{ntk}}(\beta, x_1), \cdots, f_{\mathrm{ntk}}(\beta, x_n)]^\top \in \mathbb{R}^n$ are the test data predictors. Here, $\Phi$ is the feature map corresponding to the neural tangent kernel (NTK):

$$\mathsf{K}_{\mathrm{ntk}}(x, z) = \mathbb{E}\left[\left\langle \frac{\partial f_{\mathrm{nn}}(W, x)}{\partial W}, \frac{\partial f_{\mathrm{nn}}(W, z)}{\partial W} \right\rangle\right] \tag{8}$$

where $x, z \in \mathbb{R}^d$ are any input data, and the expectation is taken over $w_r \overset{i.i.d.}{\sim} \mathcal{N}(0, I)$, $r = 1, \cdots, m$.

Under the standard assumption $\mathsf{K}_{\mathrm{ntk}}$ being positive definite, the problem Eq. (7) is a strongly convex optimization problem with the optimal predictor $u^* = \kappa^2 H^{\mathrm{cts}}(\kappa^2 H^{\mathrm{cts}} + \lambda I)^{-1} Y$ for training data, and the corresponding predictor $u^*_{\mathrm{test}} = \kappa^2 \mathsf{K}_{\mathrm{ntk}}(x_{\mathrm{test}}, X)^\top (\kappa^2 H^{\mathrm{cts}} + \lambda I)^{-1} Y$ for the test data $x_{\mathrm{test}}$, where $H^{\mathrm{cts}} \in \mathbb{R}^{n \times n}$ is the kernel matrix with $[H^{\mathrm{cts}}]_{i,j} = \mathsf{K}_{\mathrm{ntk}}(x_i, x_j)$.

We connect the problem Eq. (6) and Eq. (7) by building the following equivalence between their training and test predictors with polynomial widths:

**Theorem 3.6** (Equivalence between training neural net with regularization and kernel ridge regression for training data prediction). *Given any accuracy $\epsilon \in (0, 1/10)$ and failure probability $\delta \in (0, 1/10)$. Let multiplier $\kappa = 1$, number of iterations $T = \widetilde{O}(\frac{1}{\Lambda_0})$, network width $m \geq \widetilde{O}(\frac{n^4 d}{\Lambda_0^4 \epsilon})$ and the regularization parameter $\lambda \leq \widetilde{O}(\frac{1}{\sqrt{m}})$. Then with probability at least $1 - \delta$ over the Gaussian random initialization, we have*

$$\|u_{\mathrm{nn}}(T) - u^*\|_2 \leq \epsilon.$$

*Here $\widetilde{O}(\cdot)$ hides* $\mathrm{poly}\log(n/(\epsilon\delta\Lambda_0))$.

We can further show the equivalence between the test data predictors with the help of the multiplier $\kappa$.

**Theorem 3.7** (Equivalence between training neural net with regularization and kernel ridge regression for test data prediction). *Given any accuracy $\epsilon \in (0, 1/10)$ and failure probability $\delta \in (0, 1/10)$. Let multiplier $\kappa = \widetilde{O}(\frac{\epsilon\Lambda_0}{n})$, number of iterations $T = \widetilde{O}(\frac{1}{\kappa^2 \Lambda_0})$, network width $m \geq \widetilde{O}(\frac{n^{10} d}{\epsilon^6 \Lambda_0^{10}})$ and regularization parameter $\lambda \leq \widetilde{O}(\frac{1}{\sqrt{m}})$. Then with probability at least $1 - \delta$ over the Gaussian random initialization, we have*

$$\|u_{\mathrm{nn,test}}(T) - u^*_{\mathrm{test}}\|_2 \leq \epsilon.$$

*Here $\widetilde{O}(\cdot)$ hides* $\mathrm{poly}\log(n/(\epsilon\delta\Lambda_0))$.

### 3.2.2 Equivalence II, training with leverage scores

To apply the leverage score theory discussed in Section 3.1, Note the definition of the neural tangent kernel is exactly of the form:

$$\mathsf{K}_{\mathrm{ntk}}(x, z) = \mathbb{E}\left[\left\langle \frac{\partial f_{\mathrm{nn}}(W, x)}{\partial W}, \frac{\partial f_{\mathrm{nn}}(W, z)}{\partial W} \right\rangle\right] = \mathbb{E}_{w \sim p}[\phi(x, w)^\top \phi(z, w)]$$

where $\phi(x, w) = x\sigma'(w^\top x) \in \mathbb{R}^d$ and $p(\cdot)$ denotes the probability density function of standard Gaussian distribution $\mathcal{N}(0, I_d)$. Therefore, we try to connect the theory of training regularized neural network with leverage score sampling. Note the width of the network corresponds to the size of the feature vector in approximating the kernel. Thus, the smaller feature size given by the leverage score sampling theory helps us build a smaller upper-bound on the width of the neural nets.

Specifically, given regularization parameter $\lambda > 0$, we can define the ridge leverage function with respect to neural tangent kernel $H^{\mathrm{cts}}$ defined in Definition 3.1 as

$$q_\lambda(w) = p(w) \operatorname{tr}[\Phi(w)^\top (H^{\mathrm{cts}} + \lambda I_n)^{-1} \Phi(w)]$$

and corresponding probability density function

$$q(w) = \frac{q_\lambda(w)}{s_\lambda(H^{\mathrm{cts}})} \tag{9}$$

where $\Phi(w) = [\phi(x_1, w)^\top, \cdots, \phi(x_n, w)^\top]^\top \in \mathbb{R}^{n \times d_2}$.

We consider training the following reweighed neural network using leverage score initialization:

**Definition 3.8** (Training reweighed neural network with regularization). *Let $\kappa \in (0, 1]$ be a small multiplier. Let $\lambda \in (0, 1)$ be the regularization parameter. Let $q(\cdot) : \mathbb{R}^d \to \mathbb{R}_{>0}$ defined in* (9). *Let $p(\cdot)$ denotes the probability density function of Gaussian distribution $\mathcal{N}(0, I_d)$. We initialize the network as $a_r \overset{i.i.d.}{\sim} \mathrm{unif}[\{-1, 1\}]$ and $w_r(0) \overset{i.i.d.}{\sim} q$. Then we consider solving the following optimization problem using gradient descent:*

$$\min_W \frac{1}{2}\|Y - \kappa \overline{f}_{\mathrm{nn}}(W, X)\|_2 + \frac{1}{2}\lambda\|W\|_F^2. \tag{10}$$

*where*

$$\overline{f}_{\mathrm{nn}}(W, x) = \frac{1}{\sqrt{m}} \sum_{r=1}^m a_r \sigma(w_r^\top X)\sqrt{\frac{p(w_r(0))}{q(w_r(0))}} \text{ and } \overline{f}_{\mathrm{nn}}(W, X) = [\overline{f}_{\mathrm{nn}}(W, x_1), \cdots, \overline{f}_{\mathrm{nn}}(W, x_n)]^\top.$$

*We denote $w_r(t), r \in [m]$ as the estimate weight at iteration $t$. We denote $\overline{u}_{\mathrm{nn}}(t) = \kappa \overline{f}_{\mathrm{nn}}(W(t), X)$ as the training data predictor at iteration $t$. Given any test data $x_{\mathrm{test}} \in \mathbb{R}^d$, we denote $\overline{u}_{\mathrm{nn,test}}(t) = \kappa \overline{f}_{\mathrm{nn}}(W(t), x_{\mathrm{test}})$ as the test data predictor at iteration $t$.*

We show that training this reweighed neural net with leverage score initialization is still equivalence to the neural tangent kernel ridge regression problem (7) as in following theorem:

**Theorem 3.9** (Equivalence between training reweighed neural net with regularization and kernel ridge regression for training data prediction). *Given any accuracy $\epsilon \in (0, 1)$ and failure probability $\delta \in (0, 1/10)$. Let multiplier $\kappa = 1$, number of iterations $T = O(\frac{1}{\Lambda_0}\log(\frac{1}{\epsilon}))$, network width $m = \mathrm{poly}(\frac{1}{\Lambda_0}, n, d, \frac{1}{\epsilon}, \log(\frac{1}{\delta}))$ and regularization parameter $\lambda = \widetilde{O}(\frac{1}{\sqrt{m}})$. Then with probability at least $1 - \delta$ over the random leverage score initialization, we have*

$$\|\overline{u}_{\mathrm{nn}}(T) - u^*\|_2 \le \epsilon.$$

*Here $\widetilde{O}(\cdot)$ hides $\mathrm{poly}\log(n/(\epsilon\delta\Lambda_0))$.*

# 4 Overview of techniques

**Generalization of leverage score theory** To prove Theorem 3.3, we follow the similar proof framework as Lemma 8 in [AKM+17].

Let $K + \lambda I_n = V^\top \Sigma^2 V$ be an eigenvalue decomposition of $K + \lambda I_n$. Then conclusion (5) is equivalent to

$$\|\Sigma^{-1}V\overline{\Psi\Psi}^\top V^\top\Sigma^{-1} - \Sigma^{-1}VKV^\top\Sigma^{-1}\| \le \epsilon \tag{11}$$

Let random matrix $Y_r \in \mathbb{R}^{n \times n}$ defined as

$$Y_r := \frac{p(w_r)}{\overline{q}_\lambda(w_r)}\Sigma^{-1}V\Phi(w_r)\Phi(w_r)^\top V^\top\Sigma^{-1}.$$

where $\Phi(w) = [\phi(x_1, w), \cdots, \phi(x_n, w)]^\top \in \mathbb{R}^{n \times d_2}$. Then we have

$$\underset{\overline{q}_\lambda}{\mathbb{E}}[Y_l] = \underset{\overline{q}_\lambda}{\mathbb{E}}\left[\frac{p(w_r)}{\overline{q}_\lambda(w_r)}\Sigma^{-1}V\Phi(w_r)\overline{\Phi}(w_r)^\top V^\top\Sigma^{-1}\right] = \Sigma^{-1}VKV^\top\Sigma^{-1},$$

and

$$\frac{1}{m}\sum_{r=1}^m Y_r = \frac{1}{m}\sum_{r=1}^m \frac{p(w_r)}{\overline{q}_\lambda(w_r)}\Sigma^{-1}V\Phi(w_r)\overline{\Phi}(w_r)^\top V^\top\Sigma^{-1} = \Sigma^{-1}V\overline{\Psi\Psi}^\top V^\top\Sigma^{-1}.$$

Thus, it suffices to show that

$$\left\|\frac{1}{m}\sum_{r=1}^m Y_r - \underset{\overline{q}_\lambda}{\mathbb{E}}[Y_l]\right\| \le \epsilon \tag{12}$$

holds with probability at least $1 - \delta$, which can be shown by applying matrix concentration results. Note

$$\|Y_l\| \le s_{\overline{q}_\lambda} \text{ and } \underset{\overline{q}_\lambda}{\mathbb{E}}[Y_r^2] \preceq s_{\widetilde{q}_\lambda} \cdot \mathrm{diag}\{\lambda_1/(\lambda_1 + \lambda), \cdots, \lambda_n/(\lambda_n + \lambda)\}.$$

Applying matrix concentration Lemma 7 in [AKM+17], we complete the proof.

**Equivalence between regularized neural network and kernel ridge regression**   To establish the equivalence between training neural network with regularization and neural tangent kernel ridge regression, the key observation is that the dynamic kernel during the training is always close to the neural tangent kernel.

Specifically, given training data $x_1, \cdots, x_n \in \mathbb{R}^d$, we define the dynamic kernel matrix $H(t) \in \mathbb{R}^{n \times n}$ along training process as

$$[H(t)]_{i,j} = \left\langle \frac{\mathrm{d}f_{\mathrm{nn}}(W(t), x_i)}{\mathrm{d}W(t)}, \frac{\mathrm{d}f_{\mathrm{nn}}(W(t), x_j)}{\mathrm{d}W(t)} \right\rangle$$

Then we can show the gradient flow of training regularized neural net satisfies

$$\frac{\mathrm{d}\|u^* - u_{\mathrm{nn}}(t)\|_2^2}{\mathrm{d}t} = -2(u^* - u_{\mathrm{nn}}(t))^\top (H(t) + \lambda I)(u^* - u_{\mathrm{nn}}(t)) \tag{13}$$

$$+ 2(u_{\mathrm{nn}}(t) - u^*)^\top (H(t) - H^{\mathrm{cts}})(Y - u^*) \tag{14}$$

where term (13) is the primary term characterizing the linear convergence of $u_{\mathrm{nn}}(t)$ to $t^*$, and term (14) is the additive term that can be well controlled if $H(t)$ is sufficiently close to $H^{\mathrm{cts}}$. We argue the closeness of $H(t) \approx H^{\mathrm{cts}}$ as the consequence of the following two observations:

- Initialization phase: At the beginning of the training, $H(0)$ can be viewed as approximating the neural tangent kernel $H^{\mathrm{cts}}$ using finite dimensional random features. Note the size of these random features corresponds to the width of the neural network (scale by the data dimension $d$). Therefore, when the neural network is sufficiently wide, it is equivalent to approximate the neural tangent kernel using sufficient high dimensional feature vectors, which ensures $H(0)$ is sufficiently close to $H^{\mathrm{cts}}$.

  In the case of leverage score initialization, we further take the regularization into consideration. We use the tool of leverage score to modify the initialization distribution and corresponding network parameter, to give a smaller upper-bound of the width of the nets needed.

- Training phase: If the net is sufficiently wide, we can observe the over-parametrization phenomenon such that the weight estimate $W(t)$ at time $t$ will be sufficiently close to its initialization $W(0)$, which implies the dynamic kernel $H(t)$ being sufficiently close to $H(0)$. Due to the fact $H(0) \approx H^{\mathrm{cts}}$ argued in initialization phase, we have $H(t) \approx H^{\mathrm{cts}}$ throughout the algorithm.

Combining both observations, we are able to iteratively show the (nearly) linear convergence property of training the regularized neural net as in following lemma:

**Lemma 4.1** (Bounding kernel perturbation, informal). *For any accuracy* $\Delta \in (0, 1/10)$. *If the network width* $m = \mathrm{poly}(1/\Delta, 1/T, 1/\epsilon_{\mathrm{train}}, n, d, 1/\kappa, 1/\Lambda_0, \log(1/\delta))$ *and* $\lambda = O(\frac{1}{\sqrt{m}})$, *with probability* $1 - \delta$, *there exist* $\epsilon_W, \epsilon'_H, \epsilon'_K \in (0, \Delta)$ *that are independent of t, such that the following hold for all* $0 \leq t \leq T$:

1. $\|w_r(0) - w_r(t)\|_2 \leq \epsilon_W, \forall r \in [m]$

2. $\|H(0) - H(t)\|_2 \leq \epsilon'_H$

3. $\|u_{\mathrm{nn}}(t) - u^*\|_2^2 \leq \max\{\epsilon_{\mathrm{train}}^2, e^{-(\kappa^2 \Lambda_0 + \lambda)t/2}\|u_{\mathrm{nn}}(0) - u^*\|_2^2\}$

Given arbitrary accuracy $\epsilon \in (0, 1)$, if we choose $\epsilon_{\mathrm{train}} = \epsilon$, $T = \widetilde{O}(\frac{1}{\kappa^2 \Lambda_0})$ and $m$ sufficiently large in Lemma 4.1, then we have $\|u_{\mathrm{nn}}(t) - u^*\|_2 \leq \epsilon$, indicating the equivalence between training neural network with regularization and neural tangent kernel ridge regression for the training data predictions.

To further argue the equivalence for any given test data $x_{\mathrm{test}}$, we observe the similarity between the gradient flows of neural tangent kernel ridge regression $u_{\mathrm{ntk,test}}(t)$ and regularized neural networks $u_{\mathrm{nn,test}}(t)$ as following:

$$\frac{\mathrm{d}u_{\mathrm{ntk,test}}(t)}{\mathrm{d}t} = \kappa^2 \mathsf{K}_{\mathrm{ntk}}(x_{\mathrm{test}}, X)^\top (Y - u_{\mathrm{ntk}}(t)) - \lambda \cdot u_{\mathrm{ntk,test}}(t). \tag{15}$$

$$\frac{\mathrm{d}u_{\mathrm{nn,test}}(t)}{\mathrm{d}t} = \kappa^2 \mathsf{K}_t(x_{\mathrm{test}}, X)^\top (Y - u_{\mathrm{nn}}(t)) - \lambda \cdot u_{\mathrm{nn,test}}(t). \tag{16}$$

By choosing the multiplier $\kappa > 0$ small enough, we can bound the initial difference between these two predictors. Combining with above similarity between gradient flows, we are able to show $|u_{\text{nn,test}}(T) - u_{\text{ntk,test}}(T)| \geq \epsilon/2$ for appropriate $T > 0$. Finally, note the linear convergence property of the gradient of the kernel ridge regression, we can prove $|u_{\text{nn,test}}(T) - u^*_{\text{ntk,test}}| \geq \epsilon$.

Using the similar idea, we can also show the equivalence for test data predictors and the case of leverage score initialization. We refer to the Appendix for a detailed proof sketch and rigorous proof.

**Remark 4.2.** *Our results can be naturally extended to multi-layer ReLU deep neural networks with all parameters training together. Note the core of the connection between regularized NNs and KRR is to show the similarity between their gradient flows, as in Eq.* (15)*,* (16)*. The gradient flows consist of two terms: the first term is from normal NN training without regularizer, whose similarity has been shown in broader settings, e.g. [DZPS19, SY19, ADH$^+$19a, AZLS19b, AZLS19a]; the second term is from the $\ell_2$ regularizer, whose similarity is true for multi-layer ReLU DNNs if the regularization parameter is divided by the number of layers of parameters trained, due to the piecewise linearity of the output with respect to the training parameters.*

## 5  Conclusion

In this paper, we generalize the leverage score sampling theory for kernel approximation. We discuss the interesting application of connecting leverage score sampling and training regularized neural networks. We present two theoretical results: 1) the equivalence between the regularized neural nets and kernel ridge regression problems under the classical random Gaussian initialization for both training and test predictors; 2) the new equivalence under the leverage score initialization. We believe this work can be the starting point of future study on the use of leverage score sampling in neural network training.

## Broader Impact

The focus of this paper is purely theoretical, and thus a broader impact discussion is not applicable.

## Acknowledgments and Disclosure of Funding

Jason D. Lee acknowledges support of the ARO under MURI Award W911NF-11-1-0303, the Sloan Research Fellowship, and NSF CCF 2002272. Mengdi Wang acknowledges funding from the U.S. National Science Foundation (NSF) grant CMMI1653435, Air Force Office of Scientific Research (AFOSR) grant FA9550-19-1-020, and C3.ai DTI.

## Footnotes

[7]Strictly speaking, the optimization problem should be considered in a hypothesis space defined by the reproducing kernel Hilbert space associated with the feature/kernel. Here, we use the notation in finite dimensional space for simplicity.

[8] Our results directly extends to multi-layer deep neural networks with all layers trained together

[9] To establish the training equivalence result, we assign $\kappa = 1$ back to the normal case. For the training equivalence result, we pick $\kappa > 0$ to be a small multiplier only to shrink the initial output of the neural network. The is the same as what is used in [AKM+17].

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
