[Reviews · NeurIPS 2020]

Review 1

Summary and Contributions: The paper generalizes the result of (Arora et al 2019) on the convergence of learning with the neural tangent kernel to the regularised setting. The paper shows that training a wide enough neural net with a regularization term that depends on the 2-norm of the weights converges to learning using kernel ridge regression with the neural tangent kernel. Additionally, the paper shows that the same results hold if instead of initializing the network weights according to the normal distribution, one initializes the weights according to the leverage scores distribution of the neural network.

Strengths: Solid theoretical improvements on the convergence of learning with regularised neural networks. The paper shows that training a wide enough neural net regularized with 2-norm of the weights converges to learning using kernel ridge regression with neural tangent kernel given the width of the network is wide enough, generalizing the result of (Arora et al 2019). Furthermore, the paper shows that the same convergence results hold if instead of initializing the network weights according to the normal distribution, one initializes the weights according to the leverage scores distribution of the neural network. The connection to leverage score sampling will likely prompt future research directions in the field.

Weaknesses: The presentation is suboptimal and can be improved dramatically. There needs to be more discussion on why leverage score initialization has no improvement on the convergence compared to Gaussian initializaion.

Correctness: I checked some of the proofs quickly and did not find any issues.

Clarity: The presentation needs to be improved. More discussion is needed on weather leverage score initialization can outperform Gaussian initialization or not.

Relation to Prior Work: It is clearly discussed how this work compares to prior results.

Reproducibility: Yes

Additional Feedback: My main question is how come leverage score initialization gives no improvement on the bounds of Theorems 3.6 and 3.7. This needs discussions and good arguments. - \Lambda_0 is not quantified/defined in Theorems 3.6, 3.7, 3.9. - m \ge O(.) should become m \ge \Omega(.) in Theorems 3.6, 3.7, 3.9. - \phi(x,w) in section 3.2.2 is not correctly defined. I believe, this is a vector in R^{dm} as opposed to R^d. ============================================= I thank the authors for providing feedback. The main issue of this paper is presentation. The current version of the paper fails to convey the main technical innovations of this work. This issue has been pointed out by other reviews as well. However, I think that the presentation issue is fixable if the authors move theorem 3.6,7,9 to the beginning of the paper and make it clear that those are the central contributions of this work and discuss the prior works around this subject in more detail. On the other hand, I think that the technical contributions of this work are interesting and new, therefore I vote for acceptance.


Review 2

Summary and Contributions: This paper considers leverage score sampling to decrease the dimension of (approximate) kernel ridge regression problems. This approach has previously been used to improve the running time from O(n^3) to O(ns^2), where s is the statistical dimension of the kernel matrix. This work generalizes upon work of Avron et. al. (ICML 2017) to include a broader class of kernels. They also show that training regularized neural networks can be related to kernel ridge regression, so that their leverage score sampling techniques are then applicable.

Strengths: 1. A theoretical analysis that generalizes previous leverage score sampling to additional kernels. 2. Theoretical equivalence of *regularized* neural net initialization and kernel *ridge* regression.

Weaknesses: These results are not particularly surprising and seem slightly incremental upon the work of Avron et. al. (ICML 2017). The results would also be more convincing with implementation, but no experiments are provided in this paper.

Correctness: I did not verify the proofs but the overview of techniques (section 4) seemed reasonable. No empirical methodology is provided.

Clarity: The overview of the techniques was helpful, but the intuition of the analysis at a slightly finer granularity would have been appreciated, since the contribution of the paper is theoretical. The relation to prior work is also a bit lacking, but otherwise the paper was well-written (problem well-defined, notation clear, results are explicit, etc).

Relation to Prior Work: This work generalizes work of Avron et. al. (ICML 2017), but there is limited discussion on the results and techniques of Avron et. al. and the differences between this paper. Examples of kernels or applications handled by the approach in this paper but not by Avron et. al. would have been helpful (besides the neural nets initialization).

Reproducibility: Yes

Additional Feedback: =======Post-rebuttal update======= The author feedback states that the major contributions of the work are 1) proving the convergence result for training neural network with polynomial width, and 2) rigorously building the connection between the leverage score sampling theory and the theoretical understanding of training deep neural networks. I have to admit that I focused on the generalization of leverage score sampling to ridge leverage score sampling and largely overlooked their subsequent implications, though I also think the presentation of the paper do not reflect the importance of these contributions (e.g., the related work does not sufficiently address the results of [4,5] if neural network initialization is a major contribution of this paper).


Review 3

Summary and Contributions: This paper generalises leverage score sampling to a broader family of kernels beyond random fourier features. The paper further shows the equivalence between neural network at initialisation and random features whose dimension is proportional to the width of the network, this is a result which is know since the mid 1990s. The paper uses the above result to bring leverage score sampling to neural networks.

Strengths: This paper is a theoretical work which claims to bring leverage score sampling to the analysis of neural networks. I do not think the contributions of this paper is significant, novel and relevant enough for the Neurips community. See the comments in the next section. ------------- Post rebuttal --------------------- Sorry, I missed the contributions from this paper. Thanks the authors for the rebuttal and the other reviewers for pointing out the contributions that I have overlooked. The main contributions of this paper are: 1. Generalisation of leverage score sampling to a broader family of kernels 2. Proving the equivalence between training kernel ridge regression with neural tangent kernel and training regularised neural networks. 3. Using leverage score sampling to initialise neural network and showing that training this neural network is equivalent to training kernel ridge regression with neural tangent kernel albeit with a better bound on the network. I think these contributions make this paper a good Neurips submission but now my concern is the clarity of the paper. I find the paper difficult to read and difficult to discern the contributions. This work would have been excellent if it was well written. I encourage the authors to throughly rewrite the paper by clearly stating the contributions. I am improving the score to weak accept.

Weaknesses: I do not find any interesting novel results in this paper. The main result presented in the paper is the generalising leverage score sampling to a broader family of kernels which is not significant advancement compared to the results presented in Avron et. al ICML 2017. I do not see how we can use the results presented here to advance the understanding of neural networks. The result showing the equivalence between neural network initialisation and random features which is presented as a contribution is a result which is known since the mid 1990s [1] [2], kernel ridge regression is a special case of Gaussian processes. The result in lemma 4.1 relies on the fact that the dynamic kernel during training stays close to the kernel at initialisation which in turn is a random feature approximation of neural tangent kernel. This assumption is very strong and unrealistic. It does not apply in training real world neural networks. [1] Priors for infinite networks, Radford M. Neal [2] https://blog.smola.org/post/10572672684/the-neal-kernel-and-random-kitchen-sinks

Correctness: The claims seems to be correct. I have not checked the proof of the theorems presented in the paper as all the proofs are in the supplementary which makes the paper in itself not self contained.

Clarity: The paper is difficult to read with cryptic sentences, repeated words and notations all over the place.

Relation to Prior Work: Not very clear discussed and the paper missed an important reference - Priors for infinite networks, Radford M. Neal

Reproducibility: Yes

Additional Feedback:


Review 4

Summary and Contributions: This paper presents two lines of results: one is to generalize the study in [7] about discretizing a translation-invariant kernel with the Fourier features w^{-2 \pi i \eta \cdot x} and parameter \eta to a general form \phi(x, w) with parameter w for neural tangent kernels; and the other is to apply the result to the problem of training the connection matrix of a ReLU neural network with fixed coefficients.

Strengths: The authors first extend the study in [7] about reducing the computational complexity of kernel ridge regression by subsampling, from the Fourier features for translation-invariant kernels to a setting with general features \phi(x, w) with parameter w. This part is almost straight forward. The authors' nice contribution lies in the second part where they apply the discretization of reproducing kernels to the setting of neural tangent kernels with features \phi(x, w) being gradients of the output function in the problem of training the connection matrix of a ReLU neural network. Then they apply the error bounds for the kernel approximation to estimating the error in training the limit connection matrix. This is a novel observation, though the analysis needs to be refined to have explicit convergence bounds for the size of subsampling.

Weaknesses: The smallest eigenvalue \Lambda_0 of the sample kernel matrix H^{cts} can be very small. So the required width m can be extremely large while the regularization parameter \lambda =O(1/\sqrt{m}) is very small. When we combine the bounds in Theorem 3.3, and Theorems 3.7 and 3.9, I am not convinced that one can derive reasonable rates of convergence. The required size m for subsampling might be large and inconsistent with the motivation of reducing the complexity of kernel ridge regression. Some minor suggestions: \Lambda_0 is given only in the appendix. It needs to be specified in the text. Since the distribution for drawing the initial connection matrix is Gaussian (which is essential in the approach of neural tangent kernels), Lambda_0 should be very small when the sample size is large. The authors should give some rough estimates for \Lambda_0 and add some comments. The size m is used for both subsampling and network width, which is confusing.

Correctness: With the given tight deadline, I have not checked detailed proofs in the appendix. It seems so. But combining the bounds in Theorem 3.3, and Theorems 3.7 and 3.9 might not lead to reasonable results, as commented before.

Clarity: It is in general. But improvements can be made, as suggested before.

Relation to Prior Work: Yes.

Reproducibility: Yes

Additional Feedback:

[Author Response · NeurIPS 2020]

**Response to Reviewer #1**

• *"how come leverage score initialization gives no improvement on the bounds of Theorem 3.6 and 3.7..."*

**RE:** Thank you for the comment. The reason is that both bounds in Theorem 3.6, 3.7 come from two parts: 1) initialization phase and 2) training phase. Part 1) requires the width to be large enough, so that the initialized dynamic kernels $H(0)$ and $\hat{H}(0)$ are close enough to the neural tangent kernel (NTK) $H^{\text{cts}}$. Part 2) requires the width to be large enough, so that the dynamic kernels $H(t)$ and $\overline{H}(t)$ (defined in Definition D.3 in supplementary material) are close enough to the NTK $H^{\text{cts}}$ during the training process. Leverage score initialization optimizes the initialization bound in part 1) while keeping the bound for part 2) the same. The current state-of-art analysis gives a tighter bound in part 2), so the final bound for width is the same for both cases. If analysis for part 2) can be improved and part 1) dominates, then initializing using leverage score will be beneficial in terms of the width needed. We will address these discussion in the final version.

• *"presentation need to improve..more discussion..."* **RE:** Thank you for the advice. We will elaborate our results and address the above discussion in detail in the final version.

• *"undefined notations and typos"* **RE:** Thank you for pointing out. We will address these issues in the final version.

**Response to Reviewer #2**

• *"incremental upon the work of Avron et. al...limited discussion on differences between this paper..."*

**RE:** Thanks for your comment. We respectfully disagree. We emphasize the major contributions of our work includes 1) *proving the convergence result for training neural network with polynomial width*, and 2) rigorously *building the connection between the leverage score sampling theory and the theoretical understanding of training deep neural networks*. Both our theory of understanding training regularized neural networks and training neural networks with leverage score initialization do not exist in the literature and is not mentioned in the work of Avron et. al.. We will add more discussion about this in the final version.

• *"no experiments are provided in this paper..."*

**RE:** Thank you for the comment. This work mainly focuses on the theoretical understanding of the connection between leverage score sampling and training regularized neural networks.

**Response to Reviewer #4**

• *"equivalence between NN initialization and random feature...is known since the mid 1990s[1][2]..."*

**RE:** We respectfully disagree. We emphasize that the work in [1][2] only argues the equivalence when the hidden units in the neural network go to infinity, while our results give out specific polynomial bounds for the network width for such equivalence to hold, which is much harder. We will address this important reference in the final version.

• *"...relies on the fact that the dynamic kernel during training stays close to the kernel at initialisation which in turn is a random feature approximation of neural tangent kernel. This assumption is very strong and unrealistic..."*

**RE:** Thank you for the comment. We point out the fact that *the dynamic kernel during training stays close to the kernel at initialization* is not an assumption but a rigorously proved statement in our work. It is derived from the over-parametrization property of the neural networks, which enables the variable to converge to a point near its initialization when the network is sufficiently wide. Similar idea has shown up in previous literature [3][4] for the unregularized case.

• *"do not find any interesting novel results..."*

**RE:** Thanks for your comment. In this work, our contribution is three-folded. Apart from the generalization of leverage score sampling theory you have mentioned, we rigorously characterize the equivalence between training regularized NN and KRR by giving out a polynomial bound for the network width and number of training steps. Based upon that, we novelly apply the idea of leverage score sampling to initialize neural networks, which gives better network width bound on approximating neural tangent kernel in the initialization phase.

**Response to Reviewer #5**

• *"not convinced...derive reasonable rates of convergence"*

**RE:** This is a very good comment. The key point we claim is that the bounds we obtained in Theorem 3.6,3.7 and 3.9 on the network widths are all polynomially dependent on the size of the training dataset and the minimum eigenvalue of the neural tangent kernel. And the polynomial bounds we obtained match the state-of-art results [3][4] etc., if we set the regularization parameters to 0. We believe the constraint on the regularization parameter can be indeed relaxed by more advanced analysis, which is left as a future work.

• *"some minor suggestions..."*

**RE:** Thank you for the suggestions. We will address these notation issues and typos in the final version.

**Reference**

[1] Priors for infinite networks, Radford M. Neal

[2] https://blog.smola.org/post/10572672684/the-neal-kernel-and-random-kitchen-sinks

[3]A convergence theory for deep learning via over-parameterization, Zeyuan Allen-Zhu et. al.

[4] Gradient descent provably optimizes over-parameterized neural networks, Simon S Du et. al.

[5] On exact computation with an infinitely wide neural net, Sanjeev Arora et. al.


[Meta-Review · NeurIPS 2020]

Reviewers generally agree that the main and interesting novel contributions of the paper are in Section 3.2, where the equivalence of training neural networks with regularization and kernel ridge regression is shown with both random Gaussian and leverage score initializations. This section should be moved to the front and emphasized more. Section 3.1 on the leverage scores is rather straightforward and should not be considered a main contribution. Weakness: R5 is concerned that it might not be possible to obtain reasonable convergence rates using the bounds in Theorems 3.3, 3.7, and 3.9.